# Comparing immigration status and health patterns between Latinos and Asians: Evidence from the Survey of Income and Program Participation

Annie Ro[1]*, Jennifer Van Hook[2]

1 Department of Health, Society, and Behavior, Program in Public Health, University of California, Irvine, Irvine, CA, United States of America, 2 Department of Sociology, The Pennsylvania State University, University Park, PA, United States of America

* annie.ro@uci.edu

## Abstract

Undocumented status is widely recognized as an important social determinant of health. While undocumented immigrants have lower levels of health care access, they do not have consistently poorer physical health than the US-born or other immigrant groups. Furthermore, heterogeneity by race/ethnicity has been largely ignored in this growing literature. This paper used the 2001, 2004, 2008 panels of the restricted Survey of Income and Program Participation (SIPP), one of the only representative surveys equipped to adequately identify Asian undocumented immigrants, to compare health patterns between Asians and Latinos by immigration status. We examined three general measures of health/health access: self-rated health, disability, and current health insurance. Latino undocumented immigrants displayed some advantages in self-rated health and disability but had lower insurance coverage compared to US-born Latinos. In contrast, Asian undocumented immigrants did not differ from US-born Asians in any of the three outcomes. While undocumented status has been proposed as a fundamental cause of disease, we found no evidence that Latino and Asian undocumented immigrants consistently fare worse in health access or physical health outcomes than immigrants in other status categories. Different racial groups also appeared to have unique patterns between immigration status and health outcomes from one another.

## Introduction

Scholars are increasingly interested in how immigration status affects the life circumstances and subsequent health outcomes of the roughly 10.6 million undocumented immigrants living in the United States. Undocumented status has been described as a fundamental cause of disease that produces poorer health outcomes among undocumented immigrants compared to other groups [1]. Undocumented immigrants have clear social and economic disadvantages that are risk factors for poorer health. Median household income, one of the most robust

**Data Availability Statement:** Data can not be shared publicly because it is designated restricted use by the US Census Bureau and can only be accessed in a Federal Statistical Research Data

Center (RDC). Data are available from the US Census RDCs for researchers who meet the criteria for access to confidential data. The primary point of contact for each Federal Statistical Research Data Center is the Census Bureau employee who administers each location on behalf of all of the partner statistical agencies and RDC partner institutions. See this website for contact information for a specific RDC: https://www.census.gov/about/adrm/fsrdc/contact.html The data is third party and the authors did not receive special privileges to access the data. We received access as an approved project in the RDC.

**Funding:** AR received funding from the Russell Sage Foundation to complete this work (#93-17-05). The funders had no role in study design, data collection and analysis, decision to publish, or preparation of the manuscript. https://www.russellsage.org.

**Competing interests:** The authors have declared that no competing interests exist.

socioeconomic predictors of health status, is markedly lower among undocumented immigrants than the US-born [2]. Undocumented immigrants also face limited occupational mobility and have a higher likelihood of being employed in lower-skilled or service-sector jobs, affecting their economic resources and raising occupational health risks [3]. Being undocumented also takes an emotional toll that can affect overall health and well-being. One review found that undocumented Mexican immigrants report ubiquitous fear, stress, and depression related to their legal status [4].

The evidence supporting the idea that undocumented immigrants have poorer health outcomes is mixed, however. On one hand, undocumented immigrants have lower access to care and health insurance coverage than the US-born racial counterparts and US-born whites, as well as other documented foreign-born groups. For example, an analysis by the Migration Policy Institute found that 71% of undocumented immigrants lack health insurance compared with 40% of lawful permanent residents, 17% of naturalized citizens, and 15% of U.S.-born citizens [5]. This disparity likely reflects their lower socioeconomic status and lack of employer-based insurance, as well as their limited eligibility for public health insurance. Undocumented immigrants may also be reluctant to access institutional resources due to the fear of deportation for themselves or their families [6, 7].

Yet when examining health outcomes, the empirical evidence has not definitively established poorer physical health among undocumented immigrants compared to their documented counterparts. A recent review of 45 published studies found that the large majority did not observe significant health differences between undocumented immigrants and other immigrant groups. Some studies actually found undocumented immigrants to have a health advantage in such outcomes as blood pressure, hypertension, asthma, and self-reported chronic conditions [8]. In a nationally representative sample of farmworkers, undocumented immigrants had lower levels of chronic and acute pain than even legal permanent residents and temporary workers [8], a result the authors attribute to the "healthy immigrant" phenomenon whereby healthy people are more likely than unhealthy people to engage in risky activities such as undocumented labor migration. Consistent with this idea, a prospective study of Mexican immigrants found that compared to non-movers who remained in Mexico, undocumented Mexican immigrants had better self-rated health compared to non-movers, while documented immigrants had no difference [9].

Yet the generalizability of the literature on undocumented immigrants' health access and health status is limited by several factors. First, many previous studies had small sample sizes or did not measure immigration status directly. Analyses from nationally representative datasets with sizeable samples with accurate measurement of immigration status can help bring clarity to the patterning of health outcomes by immigration status. Furthermore, undocumented immigrants are often compared to US-born ethnic counterparts or to documented foreign-born groups with the expectation that their health outcomes should be worse. Yet immigrant incorporation scholars argue against a clear binary between undocumented and documented immigrants [8, 10, 11]. Instead, incorporation outcomes, such as health, follow a gradient whereby US-born citizens are the most privileged group with the most resources to facilitate positive outcomes, and other immigrant groups, such as citizens, legal permanent residents, or liminal statuses (e.g. temporary visa holders) have descending levels of legal security and social incorporation. Analyses that consider undocumented immigrants within the full range of immigration statuses can offer a fuller picture of health differences by varying levels of immigrant incorporation.

Finally, the majority of research examining the association between undocumented immigrants and physical health has been conducted on the aggregated population of immigrants or Latino immigrants. As a result, we know very little about potential heterogeneity across the

population, particularly by race/ethnicity. While the majority of undocumented immigrants is Latino, Asians compose a non-negligible 14% of the population–nearly 1.5 million people [12]. In comparison, immigrants from Europe and Africa comprise 5% and 2%, respectively, of the undocumented population [13]. The Asian undocumented population also shows rapid growth compared to undocumented populations from other regions of the world, notably Latin America. From 2000 to 2013, the population of undocumented immigrants from Asia increased 202% while undocumented immigrants from Mexico increased 29% [12]. In fact, since 2009, the number of undocumented immigrants from Asia continues to rise even while the total population of undocumented immigrants has plateaued [14].

We might expect differences by immigration status to be diminished among Asians compared to Latinos because their higher socioeconomic status (SES) and better health status overall [15]. The generally positive health profiles of Asian immigrants might diminish any variation in illness between documented and undocumented immigrants. Asians may also be shielded from the social consequences of being undocumented, as most negative stereotypes are applied to Latino populations [16]. If Asians do not experience the racialized stereotypes of undocumented immigrants, this can alleviate their fears of targeted deportation or engaging with formal services that benefit health, such as community health clinics.

Alternatively, the smaller population of undocumented Asian immigrants may intensify the negative impacts of undocumented status. The stigma of being undocumented in Asian communities may be higher, not only because it is rarer, but also because of cultural concerns of social desirability [17]. This elevated stigma can restrict immigrants' social networks and isolate them within their co-ethnic communities. Additionally, community and governmental organizations may provide fewer resources for the Asian undocumented because of their small population size. The lack of resources and information may leave Asian undocumented immigrants less aware about how to access services for themselves and their families. Asian undocumented immigrants also may not identify with the larger Latino undocumented community [18], removing them further from important resources and collective action.

The few studies on Asian undocumented immigrants do not adjudicate between these competing hypotheses. Qualitative studies have found undocumented Asian immigrants to report low social capital and delayed medical services because of their precarious immigration status and competing economic demands [19, 20]. One quantitative study found that Chinese immigrants underutilized health care compared to documented Chinese immigrants, but reported no difference in overall health status [21].

This project examined the association between immigration status and several health outcomes for both Asian and Latino immigrants in the Survey of Income and Program Participation (SIPP), one of the few nationally representative datasets with detailed migration information that categorizes undocumented immigrants. We first established differences in self-rated health, disability, and current health insurance coverage by different immigration statuses: US-born, naturalized foreign-born, LPR, and undocumented, among a large aggregated sample of immigrants. We expect our health outcomes to follow a gradient, with the US-born displaying the highest levels of health insurance coverage, followed by citizens, green card holders, and the undocumented immigrants with the lowest levels. For poor self-rated health and disability, we expect US-born to have the lowest levels of health outcomes, reflecting their privileges and resources. Citizens, green card holders, and undocumented immigrants will have sequentially higher levels of these outcomes. We then examined differences in these relationships by Asians and Latinos and determined the consistency of health patterns by immigration status across these two sizeable and distinct groups.

## Methods

### Data

The project received a non-human subject research determination by the UC Irvine IRB, which waives the requirement for informed consent. All data was fully anonymized before we accessed them.

We used the 2001, 2004, and 2008 panels of the Survey of Income and Program Participation. The SIPP is a longitudinal study of the U.S. civilian non-institutionalized population that collects information on income, labor force participation, social program participation, and general demographic characteristics [22]. Each panel interviews 14,000 to 52,000 households. While the SIPP is a panel study, our analysis is cross-sectional because our variables of interest were only asked at one point during the survey period. We used the restricted versions of the third and fifth modules (sixth for 2008) of each panel, as these contained the relevant migration and health information. We included respondents who self-reported being Asian or Latino (8,505 Asians; 22,795 Latinos).

The SIPP is the only nationally representative data source that includes detailed migration information that can distinguish between undocumented persons and legal non-immigrants (LNI; e.g. pre-adjustment asylees and refugees, and temporary visa holders, such as students and employees) within the population of non-LPRs (i.e. those without a green card). This migration information is limited to the restricted version of the dataset, which is only accessible in a census-designated Federal Statistical Research Data Center (FSRDC). Other representative studies, such as the California Health Interview Survey, have been previously used to study health outcomes among undocumented immigrants [23] but only contain enough migration information to identify non-LPRs. While some researchers who have used these data assume that differences between the population of non-LPRs and undocumented are minimal [23, 24], we have found considerable bias when using the non-LPRs to approximate the population of undocumented Asian immigrants. In our analyses, we dropped LNIs after separating them from the undocumented.

### Variables

**Immigration status.**   We categorized respondents into four groups: US-born, foreign-born citizen, foreign-born legal permanent resident (LPR) and undocumented. In the SIPP, all foreign-born respondents were asked about their citizenship status. Foreign-born, non-citizen respondents were further asked about their status upon arrival: immediate family or relative sponsored permanent resident, employment-based permanent resident, other permanent resident, granted refugee status or granted asylum, non-immigrants (e.g. diplomatic, student, business, or tourist visa), other. Non-citizens and non-permanent arrivals were further asked whether they have since adjusted to LPR status. We categorized immigrants who entered as refugees/asylees and non-immigrants (e.g. diplomatic, student, business, or tourist visa), have not adjusted their status, and have under 6 years US duration as LNIs, who were subsequently dropped from the analyses. We classified anyone who was not a citizen, LNI, or LPR (arrived with green card or later adjusted) as undocumented.

One concern is the level of missingness across the sequence of migration questions, which can go as high as 15%. We used Stata's mi command, which uses multiple chained equations (MICE) to estimate respondent-level missing data on all variables. This approach assumes that variables are missing at random (MAR) and that other variables in the dataset can be used to predict the missing variable. We used MICE to first handle all missing data besides migration information. We averaged across ten imputed datasets to create a single dataset that was only

missing migration information. We conducted MICE again for the migration information only and created ten datasets with imputed migration information that were combined in our analyses using Rubin's rules using Stata's mi command.

**Health outcomes.** We used three outcomes to assess general physical health. The first was *self-rated health*, which was assessed by a single item asking respondents to report their health in general as being excellent, very good, good, fair or fair. We dichotomized the measure to excellent, very good, and good (reference) versus fair and poor. The second measure was *functional limitations*. This was based on a series of questions from the adult disability module that asked about activities of daily living (ADLs), instrumental activities of daily living (IADLs), and mental, physical or sensory impairments. We followed the Census-provided coding scheme and dichotomized into severe/non severe disability versus no disability [25]. The final outcome was *current insurance*, which is part of the core module and asks whether respondents are currently insured. We dichotomized this outcome to currently insured versus not currently insured.

### Analyses

We estimated four logistic regression models for each of our health outcomes. The first examined differences in the health outcome by immigration status, adjusting only for gender, age and survey year. The second included a fuller range of covariates: race/ethnicity (Asian versus Latino), education, marital status, US region of residence, duration in the United States, and income to poverty ratio. For socioeconomic status, we chose education and income and not occupation, because of missing data for older adults and those not in the labor force. For immigration differences we included duration in the United States. Duration in the United States varies by immigration statuses due to legal provisions (e.g. green card holders are not allowed to naturalize until after five years of US residence) and there are also well-known health differences by US duration. For demographic differences, we included age, gender, marital status, and US region of residence. We included region to account for geographical differences between traditional immigration gateways and new destinations. The third model included an interaction term between race/ethnicity and immigration status to examine the modifying role of Asian ethnicity on differences by immigration status, vis-à-vis Latinos. The fourth model was the same interaction model as the third, but with a full set of covariates. We calculated predicted probabilities for our three outcomes based on the fourth model. We used Stata's mi command to combine our ten imputed datasets and accounted for the complex survey design of the SIPP by including adjusted person weights and clustering standard errors by state of residence. All analyses were conducted on Stata 14.

### Results

Table 1 shows the descriptive statistics of the sample. Among Latinos, nearly half were US-Born. Among Asians, the majority were foreign-born citizens, followed by US-born. About 8% of the Asian population and 13% of the Latino population was undocumented.

Undocumented Asian immigrants were younger than all groups and well over half of them had a college degree (62%). Undocumented Asians had the smallest proportion of employed individuals at 61% and the lowest income to poverty ratio compared to all other immigrant groups. They were distributed across all three regions of the country, whereas all other groups were concentrated in the West. Compared to the other immigrant groups, undocumented Asian immigrants were relatively recent immigrants; only 8% had over 15 years of US residence. Their health was also favorable; 76% reported having current health insurance, which was higher than LPRs. Only 5% reported having fair/poor health and 8% reported having a current disability.

**Table 1. Sample descriptives, 2001/2004/2008 Survey of Income and Program Participation.**

| | Latinos (n = 22,795) | | | | Asians (n-8,505) | | | |
|---|---|---|---|---|---|---|---|---|
| | **US-Born** | **FB Naturalized** | **FB LPR** | **FB Undoc** | **US-Born** | **FB Naturalized** | **FB LPR** | **FB Undoc** |
| | 48.4% | 17.2% | 20.9% | 13.3% | 24.2% | 44.1% | 23.4% | 7.9% |
| Mean Age | 38.0 | 46.7 | 38.2 | 32.7 | 37.7 | 47.7 | 40.1 | 33.5 |
| Male | 49.5% | 49.6% | 53.2% | 57.9% | 51.0% | 45.3% | 45.6% | 51.4% |
| Married | 44.2% | 63.1% | 57.3% | 48.7% | 44.3% | 67.5% | 65.9% | 56.8% |
| Years of Education | | | | | | | | |
| Less than HS | 21.6% | 36.1% | 54.7% | 58.5% | 6.9% | 11.6% | 15.8% | 10.3% |
| High School Graduate | 32.3% | 29.1% | 26.5% | 27.8% | 21.2% | 19.0% | 20.3% | 12.7% |
| Some College | 34.7% | 22.8% | 13.8% | 9.3% | 32.1% | 24.7% | 20.7% | 14.9% |
| College Graduate | 11.4% | 12.0% | 4.9% | 4.5% | 39.8% | 44.7% | 43.1% | 62.1% |
| Region | | | | | | | | |
| Northeast/Midwest | 26.8% | 20.5% | 17.9% | 17.7% | 24.8% | 32.1% | 37.6% | 38.6% |
| South | 35.1% | 36.2% | 36.1% | 35.1% | 12.0% | 19.2% | 22.0% | 26.3% |
| West | 38.1% | 43.4% | 46.0% | 47.2% | 62.1% | 48.6% | 40.4% | 35.1% |
| Household Size | 3.7 | 3.8 | 4.5 | 4.9 | 3.5 | 3.5 | 3.6 | 3.1 |
| Number of minors in household | 0.84 | 0.88 | 1.23 | 1.11 | 0.58 | 0.68 | 0.83 | 0.62 |
| Employed | 69.8% | 67.0% | 70.0% | 69.0% | 68.0% | 70.7% | 69.2% | 61.0% |
| Income to Poverty | 3.2 | 2.9 | 2.1 | 1.9 | 5.2 | 4.7 | 4.4 | 4.0 |
| Over 15 year US residence | | 66.2% | 41.9% | 17.0% | | 65.1% | 26.3% | 7.8% |
| Health Insurance | 72.30% | 69.0% | 42.4% | 23.1% | 84.1% | 85.8% | 71.6% | 76.1% |
| Poor Health | 14.80% | 17.0% | 12.1% | 7.7% | 7.3% | 12.1% | 9.5% | 5.4% |
| Disability | 19.60% | 19.7% | 13.2% | 8.1% | 13.3% | 18.3% | 11.6% | 7.5% |

Undocumented Latino immigrants had very different demographic features compared to Asians. While they were also younger compared to other Latino immigrant categories, Latino undocumented were majority male and unmarried. Only 5% had a college degree and 47% percent lived in the West—the largest proportion among Latino immigrant groups. The proportion employed was comparable to other groups, but they had the lowest income to poverty ratio. Only 17% had more than 15 years US duration. They had the lowest proportion of current health insurance, but also had lower prevalence of poor health and current disability compared to other Latino immigrant groups.

Table 2 shows the results of the logistic regression examining differences in fair/poor self-rated health by immigration status. Among the combined Asian and Latino sample, all groups had a lower odds of reporting fair/poor self-rated health compared to the referent US-born category. This result held after adding all covariates (Model 2) (naturalized: 0.78, p < .001; LPR: 0.71, p < .001; undocumented: 0.65, p < .001). When examining the interaction between immigration status and race, the patterns between Latinos and Asians diverged (Models 3 and 4). Compared to the US-born, only LPR Asian immigrants showed significantly higher odds of fair/poor health (Model 3, OR = 1.35, p < .05), but this difference disappeared after controlling for all covariates (Model 4). Undocumented Asian immigrants showed no significant difference from the US-born. After controlling for all covariates, all of the interaction terms for Latinos were significant, indicating there was effect modification in the self-rated health differences by race/ethnicity. Fig 1 displays the predicted probabilities of fair/poor self-rated health by immigration status for Latinos and Asians. The immigration status categories are ordered from the highest level of US membership (US-born) to the lowest (undocumented).

**Table 2. Logistic regression for immigration status and poor/fair self-rated health, main effects and interaction between race/ethnicity, 2001/2004/2008 SIPP.**

| | Model 1 –Main Effects | | Model 2- Main Effects, with cov | | Model 3- Interaction Model | | Model 4 -Interaction Model, with cov | |
|---|---|---|---|---|---|---|---|---|
| | OR | P>|t| | OR | P>|t| | OR | P>|t| | OR | P>|t| |
| **Migration Status** | | | | | | | | |
| US Born | | | | | | | | |
| Naturalized | 0.79 | ** | 0.78 | ** | 1.12 | | 1.12 | |
| LPR | 0.88 | ** | 0.71 | ** | 1.35 | * | 1.24 | |
| Undocumented | 0.76 | ** | 0.65 | ** | 1.14 | | 1.10 | |
| | | | | | | | | |
| Latino | 1.83 | ** | 1.12 | | 2.54 | ** | 1.58 | ** |
| **Status* Latino** | | | | | | | | |
| Naturalized * Latino | | | | | 0.66 | ** | 0.66 | ** |
| LPR * Latino | | | | | 0.61 | ** | 0.53 | ** |
| Undoc * Latino | | | | | 0.64 | | 0.50 | * |
| Age | 1.06 | ** | 1.06 | ** | 1.06 | ** | 1.06 | ** |
| Female gender | 1.18 | ** | 1.13 | ** | 1.17 | ** | 1.10 | ** |
| Over 15 years | | | 1.20 | * | | | 1.24 | * |
| **Education** | | | | | | | | |
| Less than HS | | | | | | | | |
| HS | | | 0.66 | ** | | | 0.66 | ** |
| Some College | | | 0.61 | ** | | | 0.60 | ** |
| BA + | | | 0.36 | ** | | | 0.36 | ** |
| **Year** | | | | | | | | |
| 2001 | | | | | | | | |
| 2004 | 0.85 | ** | 0.91 | | 0.85 | ** | 0.90 | * |
| 2008 | 0.91 | ** | 0.98 | | 0.91 | ** | 1.00 | |
| Married | | | 0.80 | ** | | | 0.77 | ** |
| **Region** | | | | | | | | |
| Northwest/Midwest | | | | | | | | |
| South | | | 0.99 | | | | 1.02 | |
| West | | | 0.91 | | | | 0.94 | |
| Income to Poverty | | | 0.84 | ** | | | 0.83 | ** |
| Intercept | 0.01 | ** | 0.03 | ** | 0.01 | ** | 0.02 | ** |

** p < .001

* p < .05.

For Latinos, the highest probability for fair/poor self-rated health was for naturalized citizens and the probability declined over the subsequent immigration categories; undocumented immigrants had the lowest probability. For Asians, undocumented immigrants similarly had a lower probability compared to the other groups, but the US-born was also lower than the other immigration categories.

Table 3 provides the results for the logistic regression models with disability as the outcome. In the combined sample of Asians and Latinos in Model 1, all immigrant groups had lower odds of functional disability compared to the US-born reference group. The same patterns held after controlling for more covariates (Model 2), and undocumented immigrants had the lowest odds (OR = 0.49, p < .01). In Model 3, when examining Asians only, LPRs were the only group to have significantly lower odds for disability compared to the US-born

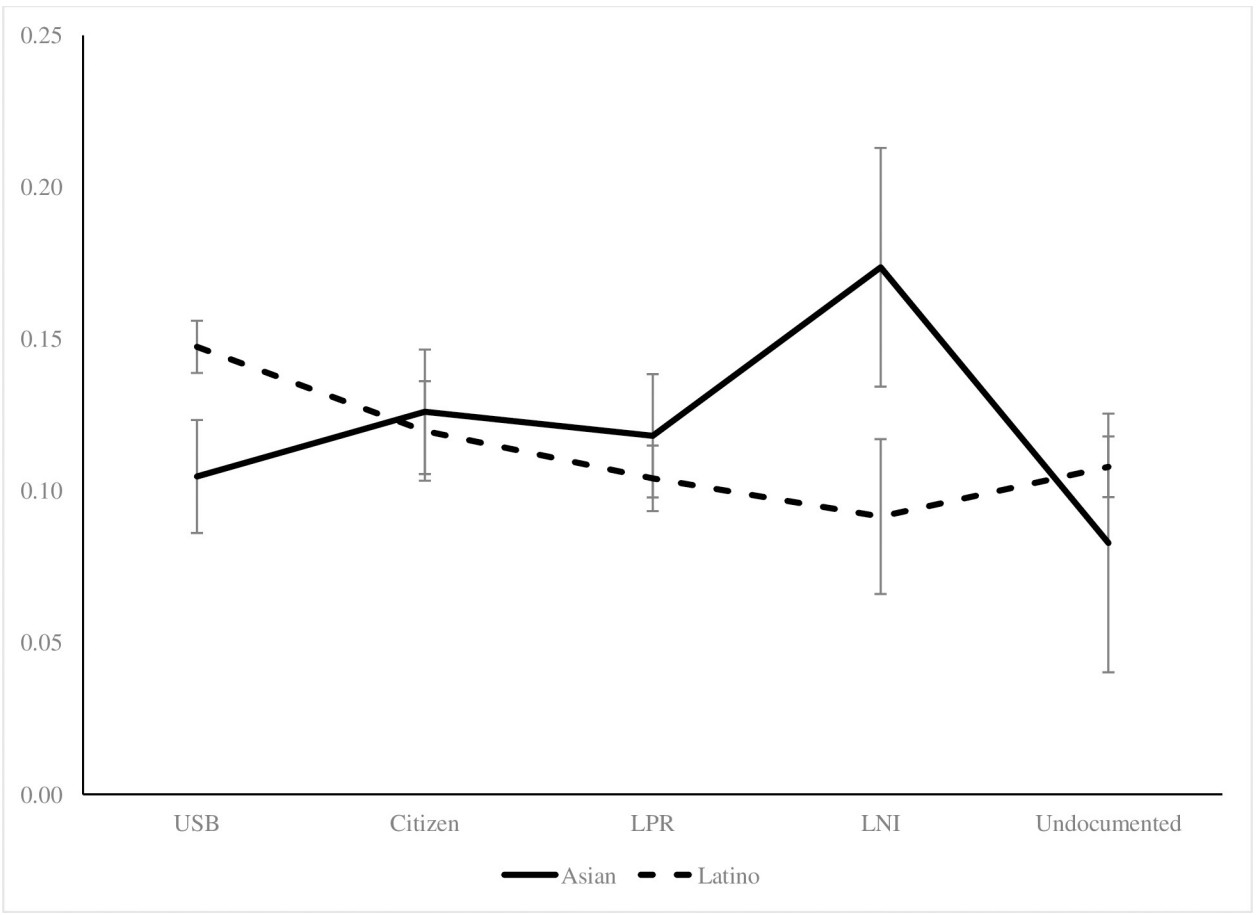

**Fig 1. Predicted probabilities of fair/poor self-rated health by immigration status, Latinos and Asians, 2001/2004/2008 SIPP.**

(OR = 0.79, p < .01). The same was true in Model 4 (OR = 0.78, p < .01). After including all covariates, interaction terms were significant and indicated bigger differences between the US-born and all immigrant groups for Latinos than Asians. Fig 2 displays the predicted probability of having a functional disability by race/ethnicity. While both groups had a downward trend for disability probability moving across immigration categories, Latinos had larger differences between undocumented immigrants and the US-born than did Asians.

Table 4 provides the results for the final outcome, current health insurance coverage. In Model 1, all foreign-born had lower odds of having current insurance compared to the US-born. After controlling for more covariates, these differences remained. In the interaction models, only the Asian LPRs had significantly lower odds of having current health insurance compared to the US-born. Asian undocumented immigrants and the US-born had statistically equivalent levels of current health insurance. After controlling for all covariates, the interaction terms for LPRs and undocumented were significant. In Fig 3, there was a clear difference between Asians and Latinos in current health insurance. For Asians, the group with the lowest current health insurance coverage was LPRs. Surprisingly, Asian undocumented immigrants had comparable probabilities with the US-born and naturalized citizens. In contrast, there was a clear decline in coverage over the immigration categories for Latinos and undocumented Latinos immigrants and undocumented immigrants had the lowest probability of having current health insurance.

**Table 3. Logistic regression for immigration status and functional disability, main effects and interaction between race/ethnicity, 2001/2004/2008 SIPP.**

| | Model 1 –Main Effects | | Model 2- Main Effects, with cov | | Model 3- Interaction Model | | Model 4 -Interaction Model, with cov | |
|---|---|---|---|---|---|---|---|---|
| | OR | P>\|t\| | OR | P>\|t\| | OR | P>\|t\| | OR | P>\|t\| |
| Migration Status | | | | | | | | |
| US Born | | | | | | | | |
| Naturalized | 0.66 | ** | 0.67 | ** | 0.88 | | 0.93 | |
| LPR | 0.64 | ** | 0.56 | ** | 0.79 | * | 0.78 | * |
| Undocumented | 0.57 | ** | 0.49 | ** | 0.78 | | 0.81 | |
| Latino | 1.39 | ** | 0.96 | | 1.76 | ** | 1.26 | ** |
| Status* Latino | | | | | | | | |
| Naturalized * Latino | | | | | 0.68 | ** | 0.66 | ** |
| LPR * Latino | | | | | 0.79 | * | 0.69 | ** |
| Undoc * Latino | | | | | 0.69 | | 0.55 | * |
| Age | 1.06 | ** | 1.06 | ** | 1.06 | ** | 1.06 | ** |
| Female gender | 1.36 | ** | 1.30 | ** | 1.35 | ** | 1.29 | ** |
| Over 15 years | | | 1.11 | | | | 1.16 | * |
| Education | | | | | | | | |
| Less than HS | | | | | | | | |
| HS | | | 0.80 | ** | | | 0.77 | ** |
| Some College | | | 0.69 | ** | | | 0.67 | ** |
| BA + | | | 0.46 | ** | | | 0.46 | ** |
| Year | | | | | | | | |
| 2001 | | | | | | | | |
| 2004 | 0.89 | | 0.93 | | 0.89 | | 0.92 | |
| 2008 | 0.87 | ** | 0.94 | | 0.87 | ** | 0.94 | |
| Married | | | 0.64 | ** | | | 0.62 | ** |
| Region | | | | | | | | |
| Northwest/Midwest | | | | | | | | |
| South | | | 0.90 | | | | 0.90 | |
| West | | | 1.00 | | | | 1.05 | |
| Income to Poverty | | | 0.90 | ** | | | 0.89 | |
| Intercept | 0.01 | ** | 0.04 | ** | 0.01 | ** | 0.03 | ** |

** p < .001

* p < .05.

## Discussion

Undocumented status is perceived as a critical factor that determines life circumstances, but physical health disadvantages among undocumented immigrants have not been consistently established in the extant literature. Furthermore, we do not know whether different racial groups show similar health patterns across different immigration categories. This is particularly true for Asians, who differ substantially in socioeconomic and health characteristics from the more-studied Latino population.

In general, we did not find undocumented immigrants to have poorer self-rated health and higher disability than their US-born counterparts. The results are consistent with others that found superior or comparable health outcomes among undocumented immigrants compared to documented immigrants [8]. Our findings join this growing body of work that finds undocumented immigrants do not have uniformly poorer health than other immigrant groups or

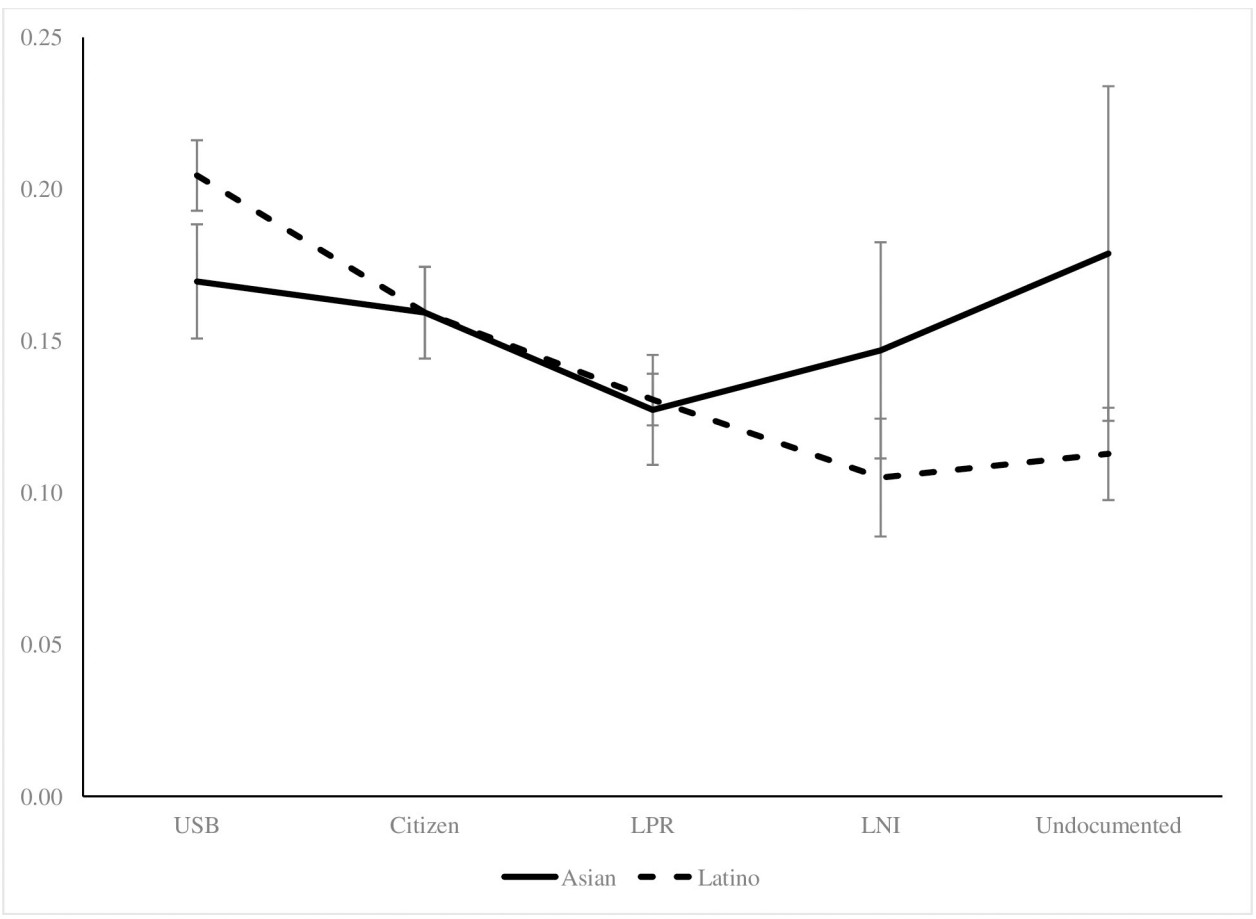

**Fig 2. Predicted probabilities of disabilities by immigration status, Latinos and Asians, 2001/2004/2008 SIPP.**

the US-born. While the findings for disability and self-rated health go against developing theory that views undocumented status as a fundamental cause of disease [26, 27], they are an extension of a well-known pattern in immigrant health generally: that the foreign-born tend to have better health than the US-born despite their lower socioeconomic status. This so-called "Immigrant Health Paradox" is attributed to protective familial and cultural behaviors, immigrant resilience, and health selection [28]. For health selection, undocumented immigrants may be even more selected than other immigrants given the costs and risks inherent in unauthorized migration, which is borne out in their superior health outcomes.

Latino undocumented immigrants displayed positive health patterns more consistently than did Asian immigrants. Among Latinos, there was a clear gradient in disability and self-rated health, such that Latino undocumented immigrants had the best health status in these outcomes compared to all other groups. The gradient was reversed for health insurance, however, and undocumented Latinos were the least likely to be insured. If we apply the health selection explanation to the self-rated health and disability patterns, it seems that undocumented immigrants are the most positively health selected, followed by green card holders and naturalized citizens. These groups also had shorter US tenure, which aligns with another well-known pattern among immigrants that recent arrivals to the United States have better health profiles than those with longer US duration [29]. Their shorter US duration means less exposure to adaptation-related stressors that may take a toll on their overall well-being (e.g. stress,

**Table 4.  Logistic regression for immigration status and current health insurance main effects and interaction between race/ethnicity, 2001/2004/2008 SIPP.**

|  | Model 1 –Main Effects | | Model 2- Main Effects, with cov | | Model 3- Interaction Model | | Model 4 -Interaction Model, with cov | |
|---|---|---|---|---|---|---|---|---|
|  | OR | P>\|t\| | OR | P>\|t\| | OR | P>\|t\| | OR | P>\|t\| |
| Migration Status |  |  |  |  |  |  |  |  |
| US Born |  |  |  |  |  |  |  |  |
| Naturalized | 0.65 | ** | 0.79 | ** | 0.84 |  | 1.00 |  |
| LPR | 0.27 | ** | 0.44 | ** | 0.41 | ** | 0.57 | ** |
| Undocumented | 0.16 | ** | 0.31 | ** | 0.64 |  | 0.99 |  |
| Latino | 0.31 | ** | 0.61 | ** | 0.47 | ** | 0.82 |  |
| Status* Latino |  |  |  |  |  |  |  |  |
| Naturalized * Latino |  |  |  |  | 0.79 | * | 0.82 |  |
| LPR * Latino |  |  |  |  | 0.64 | ** | 0.77 | * |
| Undoc * Latino |  |  |  |  | 0.20 | ** | 0.25 | ** |
| Age | 1.03 | ** | 1.03 | ** | 1.03 | ** | 1.03 | ** |
| Female gender | 1.42 | ** | 1.48 | ** | 1.42 | ** | 1.52 | ** |
| Over 15 years |  |  | 1.63 | ** |  |  | 1.61 | ** |
| Education |  |  |  |  |  |  |  |  |
| Less than HS |  |  |  |  |  |  |  |  |
| HS |  |  | 1.20 | ** |  |  | 1.21 | ** |
| Some College |  |  | 1.67 | ** |  |  | 1.67 | ** |
| BA + |  |  | 2.90 | ** |  |  | 2.57 | ** |
| Year |  |  |  |  |  |  |  |  |
| 2001 |  |  |  |  |  |  |  |  |
| 2004 | 1.13 | ** | 1.08 |  | 1.14 | ** | 1.11 | * |
| 2008 | 0.85 | ** | 0.87 | * | 0.86 | ** | 0.85 | * |
| Married |  |  | 1.55 | ** |  |  | 1.61 | ** |
| Region |  |  |  |  |  |  |  |  |
| Northwest/Midwest |  |  |  |  |  |  |  |  |
| South |  |  | 0.52 | ** |  |  | 0.55 | ** |
| West |  |  | 0.76 | ** |  |  | 0.83 | * |
| Income to Poverty |  |  | 1.26 | ** |  |  | 1.27 | ** |
| Intercept | 2.46 | ** | 0.39 | ** | 1.66 | ** | 0.27 | ** |

** p < .001

* p < .05.

dietary changes). We suspect the lower health insurance coverage among these groups reflects their higher likelihood of being employed in lower-skilled or service-sector jobs that provide fewer benefits [30] and ineligibility for public health insurance.

We did not see the same levels of positive health among Asian undocumented immigrants relative to other groups. Undocumented Asian immigrants were similar to Latinos in that they did not show any health *dis*advantages, but there were no significant differences between Asian undocumented immigrants and the US-born. While other immigrant categories (naturalized and LPR) had worse self-rated health status and lower insurance coverage than the US-born, undocumented immigrants were an exception to these trends. While the small sample size of Asian undocumented immigrants may have limited the statistical power to detect significant differences from the US-born, undocumented immigrants displayed unexpected educational advantages that may have bolstered their health relative to other groups. Although

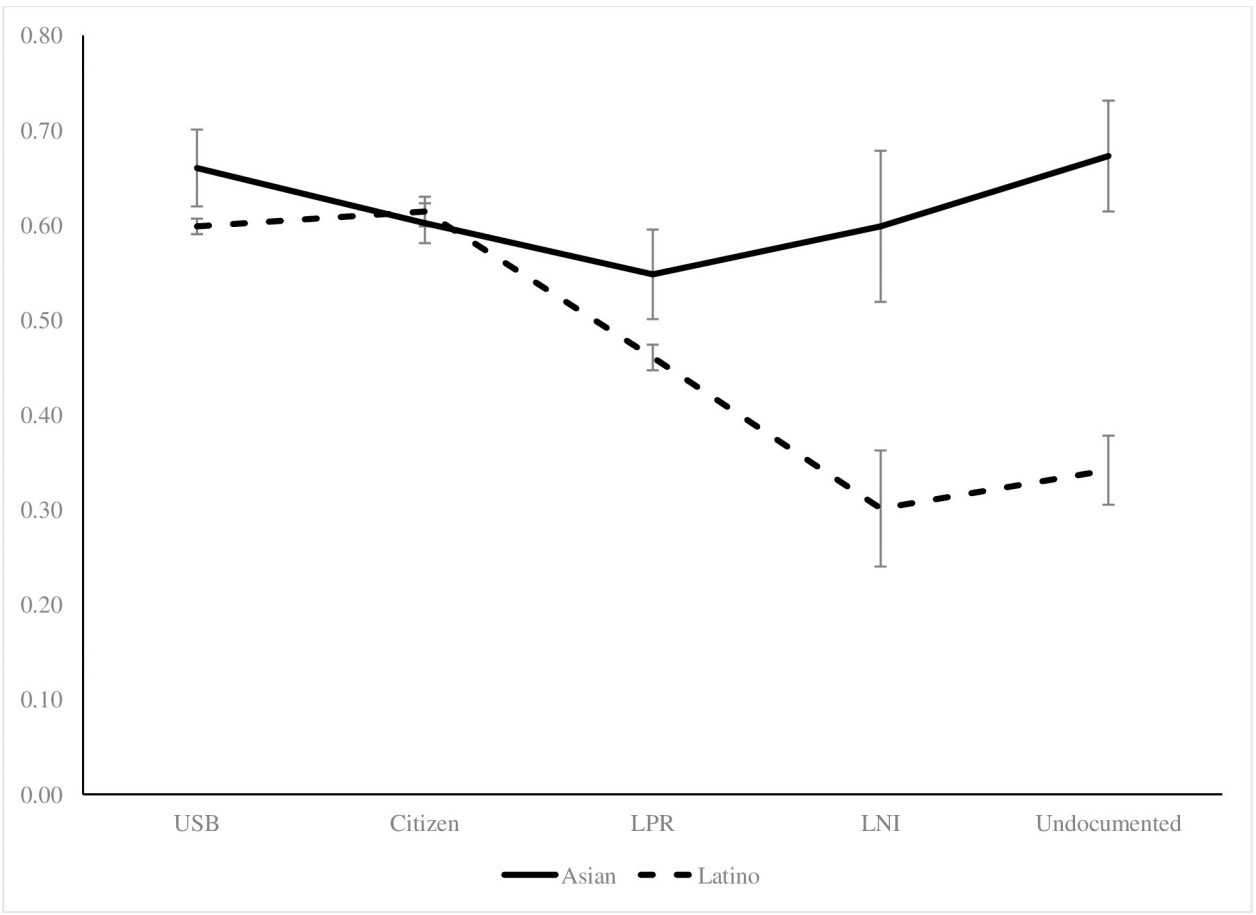

**Fig 3. Predicted probabilities of current health insurance by immigration status, Latinos and Asians, 2001/2004/2008 SIPP.**

more research is needed, the high levels of education and relatively low poverty may reflect different modes of undocumented migration among Asians compared with Latinos. A higher proportion of Asian undocumented immigrants may be migrants who overstay their tourist or student visas rather than labor migrants in search of low-skilled jobs. This mode of entry may facilitate more socioeconomic resources among undocumented immigrants and better health outcomes.

Our study also had other limitations. The small sample size of Asians precluded specific ethnic sub-group analyses. We also acknowledge that we could not categorize LNIs by specific visa type (student visa, tourist via, employment visa) so some long-term graduate students may have been categorized as undocumented based on our coding scheme. However, in the five year period before our data time period (1996–2003), student visas comprised between 3–5% of all temporary admissions from Asia [31], suggesting that long-term student visa holders are a small minority of LNIs. We also did not have data or measures to explore potential mechanisms underlying the differences we observed across immigration categories, such as health selection or acculturation. The data may not represent current trends, as much of it was collected before the heightened climate against undocumented immigrants and increased immigration enforcement activity (i.e. deportations) that began with the Obama Administration and continued through the Trump Administration. We might expect the health of undocumented immigrants to decline relative to US-born citizens if internal immigration

enforcement stressors take a toll on their well being. Nonetheless, studies from this era can inform current and future studies by providing baseline trends in the pre-enforcement, high population growth era. Finally, the SIPP oversamples low-income adults to assess their participation in federal social service programs. As a result, the demographic characteristics of our samples of undocumented immigrants are not entirely aligned with descriptive statistics from other sources, such as the American Community Survey. Yet the ACS does not directly ask about documented status and researchers use imputation techniques to estimate the likely undocumented population. In contrast, the SIPP measures immigration directly. As a result, we believe that the SIPP is the most appropriate data source for our analyses, especially for Asian immigrants.

Despite limitations, these results add to the growing literature that suggests undocumented status is not necessarily associated with poorer physical outcomes. These findings should not minimize undocumented status, but rather encourage discussion on bolstering sources of resilience that keep immigrants healthy in the face of social risk factors. Our findings also counter the narrative that undocumented immigrants drain public resources. In fact, they are healthier than the average US-born citizen. We caution, however, that the lack of health insurance among undocumented immigrants, especially among Latinos, poses a health risk in the long term. As immigrants live longer in the United States, the prevalence of poor self-rated health and disability increases, suggesting that undocumented immigrants may not have access to care when they need it. Several municipalities have implemented city or county insurance plans for undocumented immigrants ineligible for Medicaid, providing a crucial resource for medical access. Furthermore, health trends by immigration status are not consistent across racial groups, indicating differences in the experiences of being undocumented that arise from unique migration histories or available socioeconomic resources. While our findings do not align with growing theoretical discourse of undocumented status as a social risk factor, they are consistent with known immigrant health patterns, such the Healthy Immigrant Effect. Future research needs to disentangle the role of immigration status from these long-established trends in immigrant health if we wish to identify unique health risks that arise from undocumented status.

## Author Contributions

**Conceptualization:** Annie Ro, Jennifer Van Hook.

**Formal analysis:** Annie Ro.

**Funding acquisition:** Annie Ro.

**Investigation:** Annie Ro, Jennifer Van Hook.

**Methodology:** Annie Ro, Jennifer Van Hook.

**Project administration:** Annie Ro.

**Resources:** Annie Ro.

**Writing – original draft:** Annie Ro, Jennifer Van Hook.

**Writing – review & editing:** Annie Ro, Jennifer Van Hook.

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
