## [Decision Letter · Decision Letter 0]

12 Mar 2020

PONE-D-20-01253

Comparing immigration status and health patterns between Latinos and Asians: evidence from the Survey of Income and Program Participation

PLOS ONE

Dear Dr. Ro,

Thank you for submitting your manuscript to PLOS ONE. After careful consideration, we feel that it has merit but does not fully meet PLOS ONE’s publication criteria as it currently stands. Therefore, we invite you to submit a revised version of the manuscript that addresses the points raised during the review process.

We would appreciate receiving your revised manuscript by Apr 26 2020 11:59PM. To enhance the reproducibility of your results, we recommend that if applicable you deposit your laboratory protocols in protocols.io, where a protocol can be assigned its own identifier (DOI) such that it can be cited independently in the future. For instructions see: http://journals.plos.org/plosone/s/submission-guidelines#loc-laboratory-protocols

We look forward to receiving your revised manuscript.

Kind regards,

Astrid M. Kamperman

Academic Editor

PLOS ONE

Journal Requirements:

2. In ethics statement in the manuscript and in the online submission form, please provide additional information about the patient records used in your retrospective study. Specifically, please ensure that you have discussed whether all data were fully anonymized before you accessed them and/or whether the IRB or ethics committee waived the requirement for informed consent. If patients provided informed written consent to have data from their medical records used in research, please include this information.

3. Please correct your reference to "p=0.000" to "p<0.001" or as similarly appropriate, as p values cannot equal zero.

5. Your ethics statement must appear in the Methods section of your manuscript. If your ethics statement is written in any section besides the Methods, please move it to the Methods section and delete it from any other section. Please also ensure that your ethics statement is included in your manuscript, as the ethics section of your online submission will not be published alongside your manuscript.

Reviewers' comments:

Reviewer's Responses to Questions

**Comments to the Author**

1. Is the manuscript technically sound, and do the data support the conclusions?

Reviewer #1: Partly

Reviewer #2: Yes

Reviewer #3: Yes

2. Has the statistical analysis been performed appropriately and rigorously? 

Reviewer #1: Yes

Reviewer #2: Yes

Reviewer #3: Yes

3. Have the authors made all data underlying the findings in their manuscript fully available?

Reviewer #1: No

Reviewer #2: Yes

Reviewer #3: Yes

4. Is the manuscript presented in an intelligible fashion and written in standard English?

Reviewer #1: Yes

Reviewer #2: Yes

Reviewer #3: Yes

5. Review Comments to the Author

Reviewer #1: The goal of the manuscript was to compare the physical health and insurance status of U.S. Asians and Latinos by immigration status. The authors achieved their goal: they used a credible source of data that provides unique information on individuals’ immigration status. They used logistic regression to calculate the predicted probability of having fair/poor health, having any disability, and having insurance coverage by race/ethnicity and immigration status, which is suitable for their research question. They also provided a proper interpretation of their results. Overall, the manuscript presented original research findings that invite more attention to heterogeneities within the U.S. undocumented population.

However, the authors did not adequately explain some of their key analytic decisions in the paper, which can make it difficult for readers to interpret their findings. In that regard, I have three specific comments/recommendations.

It would be helpful if the authors stated their expectation about the association between the level of US membership and health / insurance coverage in the introduction section. In this section, the authors wrote about the relative disadvantage of undocumented immigrants but rarely mentioned the reference group. For example, while they were clear that undocumented immigrants have lower median household income than the U.S.-born, they were not as clear when it comes to undocumented immigrants’ “lower access to health care”. Is the expectation that the association between US membership and health / insurance is approximately linear? If so, the authors should review not just the literature on undocumented immigrants but also on naturalized citizens/ LPRs / LNIs (they briefly mentioned this in the discussion section, but it would be helpful to elaborate it in the introduction as well). Or is the expectation that undocumented status presents a distinctive challenge to health and insurance coverage, and those who are documented (U.S.-born or LNI) are expected to be homogenous? If so, the authors should reconsider their categorization of US membership.

The authors should explain how they chose the controls in their logistic regressions. The key results showed the association between race/ethnicity x immigration status and health/insurance net of not only age, gender and survey year, but also years in the U.S., education, marital status, region, and income. Why were these specific variables included? There certainly can be other observed and unobserved variables which are correlated with both race/ethnicity x immigration and health/insurance (e.g. occupation, rural/urban, health behavior…), and it is unclear whether they were excluded for a reason.

Related to the issue above, the authors sometimes set up expectations in the introduction that did not seem connected with their analysis. For example, the authors wrote: “[w]e might expect differences by immigration status to be diminished among Asians compared to Latinos because their higher socioeconomic status (SES) and better health status overall”. It seems that the point of this sentence is to establish an expectation of how undocumented Asian immigrants fare in comparison to undocumented Latino immigrants. If that’s the case, then the authors should be showing regression results not controlling for SES.

There are also a few minor issues:

- Table 1 includes some descriptives that later did not appear as regression controls: household size, number of minors in household, and employment status. The authors did not describe these numbers in the text either. I suggest that they either highlight the importance of these descriptives or delete them.

- There is a typo on line 137: “pair”

- What did the authors do about missingness on covariates? Did they also use multiple imputation or did they use a smaller sample? These could be clarified in the methods section.

Reviewer #2: This paper makes a nice contribution to the literature on immigrant health by comparing the general health status, disability status, and health insurance coverage of Latino and Asian immigrants in different legal statuses. The restricted SIPP provide the best measurement of immigrant legal status in national data and looking at this question for Asians is an important contribution. The findings -- of few differences by legal status among Asians, and of better health/lower disability risk among undocumented Latino immigrants compared to other legal statuses, while surprising, are consistent with other studies of Latino immigrants.

Several comments to strengthen/clarify the analysis:

1. it sounds like the method for determining undocumented status may miss visa overstayers, if people who enter as non-immigrants and have not adjusted their status are coded as LNIs. Is it possible to look at the years in certain statuses, especially tourist visa, to see if they have overstayed the time typically granted on the I94 (six months max, I think)?

2. Is 2% undocumented among Asians what would be estimated given the 1.5 million undocumented Asians estimate by MPI? Compare to validate the procedure for Asians. I was also surprised by the higher rate of marriage among undoc Asians, in spite of their younger age. Thoughts on this? Can you refer the reader to other studies that use the SIPP to look at undoc Asians?

3. The rates of health insurance coverage seem low to me, even for the period. 20% of USB Asians without insurance and 38% of USB Hispanics? Does your sample include children or is it limited to adults?

4. Self-rated health is well known to be biased by Spanish language. Is it possible to control for language in which the survey was answered, esp among Latinos? If not, can you speculate about how this bias may relate to legal status and the findings for SRH?

5. Even though I am familiar with other studies of this topic, I still find myself surprised by the result for health status of Latinos. Clearly there is selectivity at work: either into the sample or into the sampling universe (through in/out migration). What are the authors' thoughts on whether SIPP undoc immigrants represent all undoc immigrants? Who is not sampled, and who is not participating in the SIPP?

Minor: check line 124: non-citizens...were futher asked whether they have adjusted to LPR status. Do you mean refugees?

Reviewer #3: 1. In the abstract (line 15), it is better to use the full name of SIPP for the people who are not familiar with this dataset;

2. Is there any relationship among three dependent variables (self-rated health, disability, and current health insurance coverage)? If so, it might be helpful to explain the differences in patterns between Latino and Asian groups.

4. In line 91, you mention the order from low (undocumented) to high (US-born) which may make readers feel uncomfortable. They are different immigration statuses, but they have no ranks.

5. Can you explain more why you do cross-sectional analysis instead of longitudinal analysis since you have a longitudinal dataset? Is it related to the small sample of the undocumented Asians?

6. Can you talk more about your imputation method (line 131-134) maybe in the footnote?

7. Can you show the differences in descriptive analysis between non-missing data and missing data?

8. There are some interesting findings in your results (such as line 167-168, line 195-196), can you provide more discussion about these results in your discussion part instead of repeating the results?

9. What are the implications of your study with regards to the immigration or health policy or other community programs?

In general, it is a very clear writing paper, and the topic is important for Asian immigrants' health study.

6. PLOS authors have the option to publish the peer review history of their article (what does this mean?). If published, this will include your full peer review and any attached files.

Reviewer #1: No

Reviewer #2: No

Reviewer #3: No

---

## [Author Response · Author response to Decision Letter 0]

30 Sep 2020

We have attached our response to reviewers comments document.

---

## [Decision Letter · Decision Letter 1]

10 Nov 2020

PONE-D-20-01253R1

Comparing immigration status and health patterns between Latinos and Asians: evidence from the Survey of Income and Program Participation

PLOS ONE

Dear Dr. Ro,

Thank you for submitting your manuscript to PLOS ONE. After careful consideration, we feel that it has merit but does not fully meet PLOS ONE’s publication criteria as it currently stands. Therefore, we invite you to submit a revised version of the manuscript that addresses the points raised during the review process.

We look forward to receiving your revised manuscript.

Kind regards,

Astrid M. Kamperman

Academic Editor

PLOS ONE

Reviewers' comments:

Reviewer's Responses to Questions

**Comments to the Author**

1. If the authors have adequately addressed your comments raised in a previous round of review and you feel that this manuscript is now acceptable for publication, you may indicate that here to bypass the “Comments to the Author” section, enter your conflict of interest statement in the “Confidential to Editor” section, and submit your "Accept" recommendation.

Reviewer #1: All comments have been addressed

Reviewer #2: (No Response)

Reviewer #3: All comments have been addressed

2. Is the manuscript technically sound, and do the data support the conclusions?

Reviewer #1: Yes

Reviewer #2: Partly

Reviewer #3: Partly

3. Has the statistical analysis been performed appropriately and rigorously? 

Reviewer #1: Yes

Reviewer #2: Yes

Reviewer #3: Yes

4. Have the authors made all data underlying the findings in their manuscript fully available?

Reviewer #1: Yes

Reviewer #2: Yes

Reviewer #3: Yes

5. Is the manuscript presented in an intelligible fashion and written in standard English?

Reviewer #1: Yes

Reviewer #2: Yes

Reviewer #3: Yes

6. Review Comments to the Author

Reviewer #1: The authors have sufficiently addressed all of my comments and recommendations. Their data support their conclusions.

Reviewer #2: One argument the authors make for why their estimates differ from ACS is that the SIPP oversamples low-income households, but the authors are using survey weights to adjust for sampling, so sampling should not affect the estimates, is that correct?

Why do the authors expect a legal status gradient in health, with better outcomes for more legally privileged and vice versa, when there is no support for this in the extant literature? Is the argument that the theory is correct and the extant data/analysis is wrong? If so, then the authors should return to this in the discussion and clarify (more directly than they do currently) how their results contribute to this literature. Does positive health selection mean the theory is wrong or is that a data problem? Others have speculated and tried to consider positive health selection as a mechanism for the results -- what are next steps for this area of research in terms of disentangling this paradox? What do the authors say to people who read this and say "oh, so undoc status isn't so bad after all, look all these undoc people have great health!"

The authors addressed the concern about LNI visa overstayers by coding any LNI with more than 6years in the US as a visa overstayer, and now their undoc Asian sample is better educated than any other of the Asian groups. This seems unlikely to me: 62% of Asian undoc are college educated? I am wondering if the authors are selecting on grad students, or students who enter as BAs and then go to grad school-- a group that has fewer possibilities for adjustment of status before 6 years (as do refugees or temporary workers, who can be sponsored by their employers). Can the authors do something a little more sophisticated with the SIPP data, something akin to what is done with ACS, to impute who is an LNI and who is an overstayer among Asian LNIs with varying years of residence in the US? For instance, if the person is a grad student, foreign born, and in the US for 6+ years, don't code them as undoc? Can the authors tell in the data what kind of LNI visa a person entered with -- if tourist the timeline is different than if refugee, student, fiance, or worker. It seems that, at a minimum, the paper should present results showing sensitivity to these choices, especially since the authors want this paper to contribute by describing this heretofore unseen population (as written in the response to reviewer 3, who suggested taking out characteristics in Table 1 that do not appear in the regressions). In addition, the authors should explain these choices to the reader, as another contribution is to establish procedures for identifying and studying the Asian undoc population in survey data.

Given the quite different age profiles of the groups, and the sensitivity of health conditions, especially disability, to age, it would seem appropriate to consider whether age should be included as a non-linear term in the models.

Reviewer #3: All comments have been addressed. That would be great if you could discuss the policy context or insights from 2001 to 2008, which might be different from what we are now in the U.S. How the results from those years may inform current or further studies could be discussed more as well.

7. PLOS authors have the option to publish the peer review history of their article (what does this mean?). If published, this will include your full peer review and any attached files.

Reviewer #1: No

Reviewer #2: No

Reviewer #3: No

---

## [Editor Report · Decision Letter 2]

18 Jan 2021

Comparing immigration status and health patterns between Latinos and Asians: evidence from the Survey of Income and Program Participation

PONE-D-20-01253R2

Dear Dr. Ro,

We’re pleased to inform you that your manuscript has been judged scientifically suitable for publication and will be formally accepted for publication once it meets all outstanding technical requirements.

Kind regards,

Astrid M. Kamperman

Academic Editor

PLOS ONE
---

## [Editor Report · Acceptance letter]

22 Jan 2021

PONE-D-20-01253R2 

Comparing immigration status and health patterns between Latinos and Asians: evidence from the Survey of Income and Program Participation 

Dear Dr. Ro:

I'm pleased to inform you that your manuscript has been deemed suitable for publication in PLOS ONE. Congratulations! Your manuscript is now with our production department. 

Kind regards, 

on behalf of

Dr. Astrid M. Kamperman 

Academic Editor

PLOS ONE